# Validation of Friedewald, Martin-Hopkins and Sampson low-density lipoprotein cholesterol equations

**Gözde Ertürk Zararsız**[1,2], **Serkan Bolat**[3], **Ahu Cephe**[4©], **Necla Kochan**[5©], **Serra İlayda Yerlitaş**[1,2], **Halef Okan Doğan**[3], **Gökmen Zararsız**[1,2]*

**1** Department of Biostatistics, Erciyes University School of Medicine, Kayseri, Turkey, **2** Drug Application and Research Center (ERFARMA), Erciyes University, Kayseri, Turkey, **3** Department of Biochemistry, Sivas Cumhuriyet University School of Medicine, Sivas, Turkey, **4** Institutional Data Management and Analytics Unit, Erciyes University Rectorate, Kayseri, Turkey, **5** İzmir Biomedicine and Genome Center (IBG), İzmir, Turkey

© These authors contributed equally to this work.

* gokmenzararsiz@erciyes.edu.tr

## Abstract

### Background

Low-density lipoprotein cholesterol (LDL-C) is an important biomarker for determining cardiovascular risk and regulating lipid lowering therapy. Therefore, the accurate estimation of LDL-C concentration is essential in cardiovascular disease diagnosis and prognosis. Sampson recently proposed a new formula for the estimation of LDL-C. However, little is known regarding the validation of this formula.

### Objectives

This study aimed to validate this new formula with other well-known formulas in Turkish population, composed of adults.

### Methods

A total of 88,943 participants above 18 years old at Sivas Cumhuriyet University Hospital (Sivas, Turkey) were included to this study. LDL-C was directly measured by homogeneous assays, i.e., Roche, Beckman and Siemens and estimated by Friedewald's, Martin-Hopkins', extended Martin-Hopkins' and Sampson's formulas. The concordances between the estimations obtained by the formulas and the direct measurements were evaluated both in general and separately for the LDL-C, TG and non-HDL-C sublevels. Linear regression analysis was applied and residual error plots were generated between each estimation and direct measurement method. Coefficient of determination ($R^2$) and mean absolute deviations were also calculated.

### Results

The results showed that the extended Martin-Hopkins approach provided the most concordant results with the direct assays for LDL-C estimation. The results also showed that the

researchers who meet the criteria for access to confidential data. Data requests can be sent to a staff member in Sivas Cumhuriyet University, Faculty of Medicine and Department of Biochemistry: Demet Kablan, E-mail: demetekablan@gmail.com.

**Funding:** This study was supported by the Research Fund of Erciyes University [TSG-2021-10912]. The funders had no role in study design, data collection and analysis, decision to publish or preparation of the manuscript.

**Competing interests:** The authors have declared that no competing interests exist.

highest concordances were obtained between the direct assays with the extended Martin-Hopkins formula calculated with the median statistics obtained from our own population. On the other hand, it was observed that the results of the methods may differ in different assays. The extended Martin-Hopkins approach, calculated from the median statistics of our population, gave the most concordant results in patients with "low LDL-C level (LDL-C levels < 70 mg/dL) or hypertriglyceridemia (TG levels ≥ 400 mg/dL)".

## Conclusions

Although the results of the formulas in different assays may vary, the extended Martin-Hopkins approach was the best one with the highest overall concordances. The validity of the Martin Hopkins' and Sampson's formulas has to be further investigated in different populations.

## Introduction

Cardiovascular diseases (CVDs) are the leading causes of death worldwide which accounts for 32% of all global deaths in 2019 [1]. There are many different risk factors that increase the likelihood of developing CVD such as smoking, unhealthy diet, obesity, physical inactivity and excessive use of alcohol consumption. However, elevated low-density lipoprotein cholesterol (LDL-C) concentration is the major risk factor associated with an increased risk of CVD mortality. Clinical studies have shown that there exists a strong and positive correlation between LDL-C concentration and the development and progression of CVD [2, 3]. Hence, LDL-C is a major determinant which is used as a target measure in clinical practice guidelines and to investigate appropriate treatment strategies.

The gold standard for measuring LDL-C level is $\beta$-quantification which combines ultracentrifugation and precipitation with poly-anions in order to separate lipoprotein particles [4]. However, $\beta$-quantification, is not convenient for routine use since it is expensive, time consuming and requires a large number of sample batches and other instruments [2, 3, 5, 6]. Therefore, the use of this measuring method is limited to a few specialized laboratories [7]. In 1972, a new method called LDL-C estimation by Friedewald formula was introduced and it has become a new standard in clinical practice guidelines worldwide due to its advantages such as convenient, cost-effective and time-saving compared to the direct method, LDL-C with ultracentrifugation followed by $\beta$-quantification [2]. Although it has been widely adopted in clinical practice there are some limitations to this method. Firstly, division of TG directly by a fixed factor of 5, a fixed ratio of TG:VLDL-C, does not provide an accurate estimate for VLDL-C. Secondly, the Friedewald formula requires fasting serum to accurately estimate LDL-C since chylomicronemia in a non-fasting situation leads to the overestimation of VLDL-C [8]. It is also inappropriate to use Friedewald equation in the presence of high TG concentration (TG ≥ 400 mg/dL) and dysbetalipoproteinaemia. Thirdly, the LDL-C levels < 70 mg/dL and TG ≥ 150 mg/dL underestimates LDL-C levels and this may result in undertreatment of the patients [9].

To overcome the aforementioned limitations, Martin et al. proposed a novel equation to accurately estimate LDL-C levels. This new equation is called Martin-Hopkins formula which uses an adjustable factor for TG:VLDL-C ratio based on TG and non-high-density lipoprotein cholesterol (non-HDL-C) levels [10]. This formula was validated using a large sample of lipid

profiles and it was shown in their study that the Martin-Hopkins formula provides more accurate LDL-C estimates compared to Friedewald formula, especially in patients with LDL-C < 70 mg/dL and those with elevated TG levels [10]. However, despite very promising results, this method did not reach widespread use. Since then, many studies have been conducted to demonstrate the validity of Martin-Hopkins formula, which results in more accurate estimates of LDL-C concentration particularly in TG < 400 mg/dL compared to Friedewald formula [11–14]. Due to the advantages of Martin-Hopkins formula, in 2018 the new American College of Cardiology and American Heart Association guidelines on CVD risk biomarkers recommended the use of Martin-Hopkins method as the preferred estimation method for low LDL-C individuals [15].

Since Friedewald and Martin-Hopkins formulas are developed and validated for patients with TG < 400 mg/dL, clinical laboratories generally do not report LDL-C levels for hypertriglyceridemia patients whose TG levels > 400 mg/dL. Chylomicrons accumulate at high TG levels and may change the association between TG and VLDL-C. Therefore, Friedewald formula causes larger errors in LDL-C estimate. On the other hand, Martin-Hopkins formula includes an adjustable factor parameter for splitting TG levels into categories, however it has never been validated for patients with TG levels above 400 mg/dL [10, 12, 16]. Therefore, clinical laboratories perform direct chemical assays in order to measure the LDL-C levels at higher TG levels, but these direct assays lack standardization, are time consuming and costly [17]. In 2020, with an aim of estimating LDL-C levels for patients with TG levels up to 800 mg/dL, Sampson et al. derived a new novel equation (hereafter referred to as the Sampson formula) which uses $\beta$-quantification results obtained out of a population with a high frequency of hypertriglyceridemia [18]. The authors claimed that the new formula not only enables clinicians to report LDL-C levels for patients with hypertriglyceridemia (TG level $\leq$ 800 mg/dL), but also estimates LDL-C levels for patients with normopolidemia and/or low level of LDL-C the same as or more accurate than other existing equations [18]. Recently, Sajja et al. performed an extended Martin-Hopkins formula and compared its accuracy with Sampson's formula [19]. This extended formula uses strata-specific median ratio of TGs:VLDL-C to estimate LDL-C levels for patients with TG levels between 400 and 799mg/dL. Their results showed that the extended Martin-Hopkins formula gives a more accurate estimate compared to Friedewald and Sampson formulas at TG levels of 400 to 799 mg/dL and also performs better at low LDL-C levels [19].

In this study, we used a very large sample size and evaluated the validity of the LDL-C levels estimated by Friedewald, Martin-Hopkins, extended Martin-Hopkins and Sampson formulas with the LDL-C levels measured by some direct assays (i.e., Roche, Beckman and Siemens) using the Turkish population.

## Materials and methods

### Study population

A total of 88,943 samples were included the study. The demographic characteristics of the participants according to analytical platforms were given in Table 1. We reviewed the levels of the HDL-C, LDL-C, triglycerides, and total cholesterol in these samples. Data were obtained by the Sivas Cumhuriyet University Medical Faculty, Department of Biochemistry from March 3, 2011, to December 31, 2019. These parameters are ordered from a wide variety of clinical units in the Medical Faculty. The study was approved by the local ethics committee in accordance with the Declaration of Helsinki (2020–03/05). Since the study was designed as retrospective, no informed consent was obtained from participants. We did not categorize the participants based on fasting status.

**Table 1. Study population characteristics.**

| Characteristic | Overall (*N* = 88, 943) | Roche (*N* = 39, 558) | Beckman (*N* = 30, 087) | Siemens (*N* = 19, 298) |
|---|---|---|---|---|
| Age (years) | 53.04±17.64 | 52.85±17.71 | 52.61±17.69 | 54.08±17.39 |
| Gender | | | | |
| Female | 46,246 (52.0) | 21,186 (53.6) | 15,614 (51.9) | 9,446 (48.9) |
| Male | 42,697 (48.0) | 18,372 (46.3) | 14,473 (48.1) | 9,852 (51.1) |
| Lipid values | | | | |
| TC (mg/dL) | 181 (150–213) | 177 (147–208) | 191 (160–226) | 173 (144–205) |
| TG (mg/dL) | 130 (92–188) | 130 (93–188) | 128 (90–186) | 131 (92–192) |
| HDL-C (mg/dL) | 42 (35–51) | 42 (35–52) | 44 (37–52) | 40 (33–49) |
| Non-HDL-C (mg/dL) | 136 (108–167) | 132 (104–162) | 146 (117–178) | 131 (104–160) |
| TG—TC ratio | 0.74 (0.54–1.03) | 0.76 (0.56–1.04) | 0.68 (0.50-.96) | 0.78 (0.56–1.10) |
| LDL-C$_D$ (mg/dL) | 114 (88–141) | 112 (87–139) | 127 (102–152) | 98 (75–123) |
| LDL-C$_F$ (mg/dL) | 106 (81–133) | 102 (78–128) | 116 (91–144) | 101 (77–126) |
| LDL-C$_S$ (mg/dL) | 109 (85–136) | 105 (81–131) | 119 (94–147) | 104 (80–129) |
| LDL-C$_M$ (mg/dL) | 110 (86–137) | 106 (82–132) | 120 (95–147) | 105 (82–131) |
| LDL-C$_E$ (mg/dL) | 115 (90–141) | 113 (88–139) | 129 (104–154) | 99 (77–123) |

Values are expressed as *N*(%), mean±SD or median(1$^{st}$—3$^{rd}$ quartiles). TC: total cholesterol; TG: triglycerides; HDL-C: high-density lipoprotein cholesterol; LDL-C: low-density lipoprotein cholesterol; Non-HDL-C: non-high-density lipoprotein cholesterol; LDL-C$_D$: LDL-C measured by direct assay; LDL-C$_F$: LDL-C calculated by Friedewald formula; LDL-C$_S$: LDL-C calculated by Sampson formula; LDL-C$_M$: LDL-C calculated by Martin-Hopkins formula; LDL-C$_E$: LDL-C calculated by the extended Martin-Hopkins formula.

## Lipid measurements

Different systems were used to directly measure HDL-C, LDL-C, triglycerides, and total cholesterol parameters. Detailed measurement procedures were given below according to used systems.

*Roche Cobas 8000, c-702 and c-501*: Total cholesterol, triglycerides and LDL-C, HDL-C measurements were performed using colorimetric enzymatic reaction.

*Siemens Advia 1800*: HDL-C levels were determined with Trinder reaction. Triglycerides and LDL-C, HDL-C measurements were performed using colorimetric enzymatic reaction.

*Beckman Coulter AU5800*: Total cholesterol, triglycerides and LDL-C, HDL-C measurements were performed using colorimetric enzymatic reaction.

## Lipid estimations

Direct measurement of LDL-C (LDL-C$_D$) was measured directly with one of the Roche, Beckman, or Siemens assays. Friedewald's LDL-C estimation (LDL-C$_F$) was calculated with the following Friedewald formula [2]:

$$\text{LDL-C}_F = \text{TC} - \text{HDL-C} - (\text{TG}/5) \tag{1}$$

Sampson's LDL-C estimation (LDL-C$_S$) was calculated using the least squares formula mentioned by Sampson [18]:

$$\text{LDL-C}_S = \frac{\text{TC}}{0.948} - \frac{\text{HDL-C}}{0.971} - \left( \frac{\text{TG}}{8.56} + \frac{\text{TG} \times \text{non-HDL-C}}{2140} - \frac{\text{TG}^2}{16100} \right) - 9.44 \tag{2}$$

Martin-Hopkins and the extended Martin-Hopkins LDL-C estimations (LDL-C$_M$ and LDL-C$_E$, respectively) were calculated using the Martin-Hopkins formula [10]:

$$LDL\text{-}C_M = TC - HDL\text{-}C - (TG/\zeta) \tag{3}$$

In this formula, $\zeta$ is an adjustable factor and was calculated using the median TG/VLDL-C ratio, which takes into account the sublevels of TG and non-HDL-C levels. For LDL-C$_M$ estimation, $\zeta$ was obtained from the strata specific median ratio of TG/VLDL-C in the 180-cell table suggested by [10]. For LDL-C$_E$, $\zeta$ values were estimated using Turkish population data. For this purpose, two-dimensional tables were created and strata specific median ratio of TG/VLDL-C were calculated from different combinations by using the accepted cut-off values for TG and non-HDL-C levels. Tables containing the median ratio of TG/VLDL-C with 30, 70, 130, 180, 420 and 780 cells were generated, including 5 and 30 sublevels for TG and 6, 14 and 26 sublevels for non-HDL-C (S1 File). Overall concordances were calculated for each combination. The most concordant results for the Roche and Siemens assays, and the second most concordant results for Beckman assay were obtained from the coefficients in the 180-cell tables. Although the Beckman direct assay yielded the most concordant results in 420-cell tables, the overall concordance was only 0.14% higher compared to calculations based on 180-cell tables (Fig 1). For this reason, the 180-cell tables were used to calculate the LDL-C$_E$ estimations in all of the assays.

## Statistical analysis

Overall concordances of LDL-C estimates were calculated for each assay separately. Overall concordance was defined as the ratio of direct LDL-C (LDL-C$_D$) in the same category as estimated LDL-C based on estimated LDL-C levels (< 70 mg/dL, 70 to 99 mg/dL, 100 to 129 mg/dL, 130 to 159 mg/dL, 160 to 189 mg/dL and ≥ 190 mg/dL). In addition, overall concordances for LDL-C estimates were also calculated for the TG and non-HDL-C sublevels. Ordinary least squares linear regression analyses were conducted to compare the estimated and measured

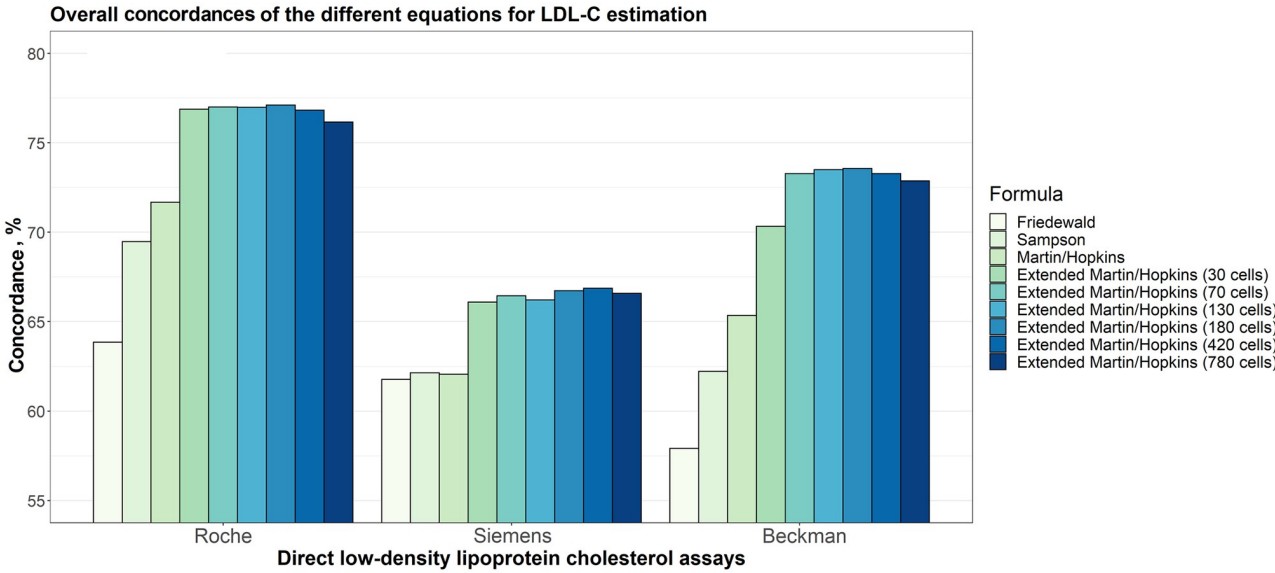

**Fig 1. Overall concordances of the different equations for LDL-C estimation.** Overall concordances of different equations for LDL-C estimation for each assay (i.e., Roche, Siemens and Beckman) are given in a clustered bar chart. Each bar indicates the concordance of estimating LDL-C levels by different formulas given that the LDL-C levels measured by Roche, Siemens and Beckman direct methods, accordingly.

LDL-C values. Residual error plots were also generated from the difference of each LDL-C estimation method and direct LDL-C measurements according to TG levels. All analyses were conducted using R 4.0.4 (www.r-project.org) statistical software.

## Results

### Patient characteristics

Demographic details of all participants are provided in Table 1. In this study, a total of 88,943 profiles were taken, out of this 52% were females and 48% were males. The mean age of all participants was 53.04 ± 17.64. The median values of LDL-C measured by direct method, TC, TG, and HDL-C levels were 114 mg/dL, 181 mg/dL, 130 mg/dL, and 42 mg/dL, respectively. The median of nonHDL-C level, and TG/TC ratio were calculated as 136 mg/dL and 0.74 mg/dL, respectively. The demographic details of participants whose HDL-C, LDL-C, triglycerides and total cholesterol parameters are measured by Roche ($N$ = 39, 558), Beckman ($N$ = 30, 087) and Siemens ($N$ = 19, 298) direct assays are also given in Table 1, separately.

### Overall concordances of the different equations for LDL-C estimation

In order to estimate LDL-C levels using Martin-Hopkins' formula, strata specific median ratio of TG/VLDL-C were calculated from different combinations by using the accepted cut-off values for TG and non-HDL-C levels. To this end, two-dimensional tables containing the median ratio of TG/VLDL-C with 30, 70, 130, 180, 420 and 780 cells were generated, including 5 and 30 sublevels for TG and 6, 14 and 26 sublevels for non-HDL-C.

Overall concordances of LDL-C estimates for each assay are given in Fig 1. It can be seen from the figure that the overall concordances of the different equations implemented in this study to estimate LDL-C levels for Siemens direct method are lower than those with other two direct assays: Roche and Beckman. It can also be seen that the extended Martin-Hopkins with different number of cells produced the highest concordances for each assay. Even for tables with 30 cells, the concordance levels for Roche and Siemens were similar with tables with cells greater than 30. Surprisingly, the most concordant results were obtained when extended Martin-Hopkins formula with 180-cell table was performed for Roche and Beckman direct assays. In Siemens direct assay, although the highest concordance was obtained for the 420-cell table, there was a very slight increase in the concordance (0.14%) compared to the 180-cell table. Thus, we focused on 180-cell table for the extended Martin-Hopkins formula for all assays. The 180-cell table, which includes the median statistics for TG/VLDL-C ratio is given in Table 2 and the other combinations are given in S1 File.

### Distribution density of LDL-C concentrations calculated by direct methods and different formulas

The raincloud plots of measured LDL-C levels by direct methods and LDL-C estimates using the formulas considered in this study are provided by Fig 2. The red line in this figure shows the difference between the median of measured LDL-C with Roche, Beckman or Siemens and the median of estimated LDL-C levels. It is evident from the figure that all of the formulas except extended Martin-Hopkins formula underestimated the LDL-C levels when Roche and Beckman assays were used, whereas all formulas except Friedewald's formula overestimated the LDL-C levels when Siemens direct assay was utilized. On the other hand, the distribution pattern for this assay was the most similar, with Martin-Hopkins formula.

**Table 2. Median statistics for the ratio of triglycerides to very low-density lipoprotein cholesterol by the cross table of non-high-density lipoprotein cholesterol and triglycerides calculated from the Turkish population (calculated for each direct assay method for 180-cell strata).**

| TG Levels (mg/dL) | Non-HDL-C (mg/dL) | | | | | |
|---|---|---|---|---|---|---|
| | <100 | 100–129 | 130–159 | 160–189 | 190–219 | ≥220 |
| 7–49 | R:6.00 B:6.82 S:2.34 | R:5.22 B:4.04 S:2.10 | R:3.50 B:3.91 S:1.50 | R:4.43 B:2.01 S:1.21 | R:2.11 B:2.07 S:0.62 | R:2.11 B:1.02 S:0.62 |
| 50–56 | R:7.00 B:8.17 S:3.11 | R:5.60 B:5.00 S:2.47 | R:4.23 B:3.50 S:1.92 | R:3.93 B:2.24 S:2.15 | R:2.23 B:2.83 S:0.98 | R:3.65 B:1.98 S:0.39 |
| 57–61 | R:7.13 B:9.50 S:3.20 | R:6.67 B:6.10 S:2.73 | R:4.83 B:4.46 S:2.32 | R:3.69 B:3.63 S:2.63 | R:2.35 B:2.68 S:1.21 | R:2.35 B:3.05 S:0.99 |
| 62–66 | R:7.22 B:9.88 S:3.37 | R:7.11 B:6.40 S:2.52 | R:5.73 B:4.77 S:2.17 | R:4.74 B:2.71 S:2.46 | R:4.40 B:2.54 S:1.41 | R:12.80 B:1.99 S:1.41 |
| 67–71 | R:7.67 B:9.57 S:3.68 | R:6.41 B:7.05 S:2.84 | R:5.38 B:4.63 S:2.53 | R:5.00 B:3.55 S:2.18 | R:2.63 B:2.65 S:2.92 | R:2.09 B:1.43 S:2.92 |
| 72–75 | R:7.20 B:9.25 S:3.75 | R:6.82 B:7.40 S:3.13 | R:5.73 B:4.93 S:2.71 | R:5.14 B:4.00 S:2.03 | R:4.55 B:2.64 S:1.77 | R:2.62 B:2.11 S:1.77 |
| 76–79 | R:7.80 B:9.75 S:3.66 | R:7.05 B:7.70 S:3.14 | R:5.64 B:5.64 S:3.16 | R:5.43 B:3.71 S:2.71 | R:2.97 B:2.75 S:2.35 | R:2.81 B:2.60 S:0.80 |
| 80–83 | R:7.36 B:13.33 S:3.91 | R:6.88 B:7.77 S:3.47 | R:5.86 B:5.19 S:2.89 | R:5.63 B:4.37 S:3.20 | R:4.21 B:2.85 S:1.74 | R:2.65 B:2.35 S:2.37 |
| 84–87 | R:7.82 B:10.88 S:4.25 | R:7.82 B:7.17 S:3.41 | R:6.00 B:5.67 S:3.26 | R:5.59 B:3.58 S:2.29 | R:5.12 B:3.19 S:1.49 | R:8.40 B:2.97 S:1.76 |
| 88–92 | R:8.00 B:12.04 S:4.09 | R:8.18 B:7.42 S:3.83 | R:6.13 B:5.93 S:3.54 | R:5.08 B:4.74 S:2.73 | R:4.05 B:3.37 S:1.97 | R:2.94 B:2.49 S:1.75 |
| 93–96 | R:8.55 B:10.56 S:4.43 | R:7.15 B:8.00 S:3.96 | R:6.52 B:5.88 S:3.43 | R:5.53 B:4.36 S:3.29 | R:6.33 B:3.72 S:3.77 | R:4.89 B:3.47 S:1.08 |
| 97–100 | R:7.54 B:12.38 S:4.25 | R:7.69 B:8.82 S:3.90 | R:6.96 B:6.53 S:3.05 | R:6.57 B:4.81 S:3.05 | R:5.00 B:3.59 S:2.19 | R:6.19 B:2.56 S:1.95 |
| 101–105 | R:7.92 B:11.44 S:4.81 | R:8.00 B:8.58 S:4.13 | R:7.11 B:6.44 S:3.68 | R:6.50 B:5.00 S:2.94 | R:6.71 B:4.21 S:3.76 | R:3.74 B:2.81 S:2.25 |
| 106–110 | R:8.27 B:12.06 S:4.91 | R:7.33 B:9.08 S:4.09 | R:7.64 B:6.78 S:3.53 | R:6.47 B:5.24 S:2.69 | R:5.40 B:3.79 S:2.69 | R:5.00 B:3.03 S:3.72 |
| 111–115 | R:8.14 B:12.33 S:5.14 | R:7.67 B:9.25 S:4.63 | R:7.96 B:7.13 S:3.99 | R:6.74 B:4.87 S:3.35 | R:5.23 B:4.11 S:3.29 | R:4.48 B:3.02 S:3.06 |
| 116–120 | R:7.73 B:12.00 S:5.29 | R:7.87 B:9.19 S:4.33 | R:7.44 B:6.82 S:4.20 | R:7.00 B:5.27 S:3.55 | R:5.65 B:4.46 S:3.53 | R:10.90 B:3.08 S:3.24 |
| 121–126 | R:8.40 B:15.75 S:5.30 | R:8.13 B:10.25 S:4.31 | R:7.35 B:7.69 S:4.25 | R:7.09 B:5.70 S:3.73 | R:6.15 B:4.54 S:2.80 | R:5.59 B:3.97 S:2.41 |
| 127–132 | R:8.57 B:15.94 S:5.30 | R:8.19 B:9.77 S:4.45 | R:7.53 B:7.94 S:4.21 | R:7.19 B:5.52 S:3.78 | R:6.19 B:4.40 S:3.12 | R:5.33 B:3.49 S:3.11 |
| 133–138 | R:7.75 B:14.78 S:5.04 | R:8.59 B:10.42 S:4.74 | R:8.06 B:7.44 S:4.15 | R:6.90 B:6.11 S:3.79 | R:5.90 B:5.06 S:3.45 | R:6.18 B:3.62 S:2.84 |
| 139–146 | R:8.26 B:16.06 S:5.33 | R:8.75 B:10.21 S:4.71 | R:7.94 B:8.29 S:4.77 | R:7.28 B:5.88 S:4.25 | R:7.42 B:5.15 S:3.47 | R:5.32 B:3.75 S:2.90 |
| 147–154 | R:7.84 B:15.92 S:5.52 | R:8.82 B:11.38 S:5.07 | R:8.28 B:8.44 S:4.61 | R:7.45 B:6.67 S:3.94 | R:6.61 B:4.78 S:3.75 | R:5.65 B:3.76 S:3.20 |
| 155–163 | R:8.00 B:17.44 S:6.00 | R:8.37 B:11,43 S:4.89 | R:7.93 B:9.53 S:4.88 | R:7.80 B:6.87 S:4.30 | R:6.87 B:5.61 S:4.03 | R:5.59 B:4.45 S:3.25 |
| 164–173 | R:7.86 B:15.41 S:5.90 | R:8.38 B:12.69 S:5.44 | R:8.05 B:9.88 S:4.86 | R:7.59 B:7.26 S:4.29 | R:6.58 B:5.39 S:3.98 | R:5.43 B:5.34 S:3.38 |
| 174–185 | R:7.87 B:16.27 S:6.79 | R:8.32 B:12.71 S:5.68 | R:8.76 B:9.89 S:5.00 | R:8.02 B:7.36 S:4.38 | R:7.50 B:6.07 S:3.87 | R:5.76 B:4.14 S:3.69 |
| 186–201 | R:8.64 B:18.09 S:6.19 | R:8.59 B:12.87 S:5.89 | R:8.38 B:10.34 S:5.37 | R:7.72 B:8.78 S:4.75 | R:7.15 B:6.46 S:4.12 | R:5.68 B:5.03 S:3.58 |
| 202–220 | R:8.15 B:18.07 S:6.11 | R:8.76 B:14.71 S:6.34 | R:8.83 B:11.16 S:5.63 | R:8.20 B:8.46 S:5.09 | R:7.59 B:6.66 S:4.69 | R:6.77 B:5.81 S:4.40 |
| 221–247 | R:8.81 B:21.27 S:6.94 | R:8.79 B:15.77 S:6.47 | R:8.67 B:11.60 S:5.70 | R:8.23 B:9.04 S:5.00 | R:7.44 B:7.09 S:4.92 | R:7.00 B:5.66 S:4.42 |
| 248–292 | R:8.56 B:20.77 S:7.14 | R:8.66 B:15.78 S:6.49 | R:8.83 B:12.13 S:5.82 | R:8.23 B:10.00 S:5.37 | R:7.85 B:8.29 S:5.02 | R:7.15 B:6.16 S:4.40 |
| 293–399 | R:8.76 B:24.86 S:7.82 | R:8.69 B:19.15 S:7.09 | R:8.68 B:14.77 S:6.48 | R:8.05 B:11.44 S:5.79 | R:7.75 B:9.45 S:5.25 | R:7.15 B:6.83 S:4.56 |
| ≥400 | R:12.16 B:23.56 S:8.83 | R:8.54 B:17.93 S:7.45 | R:8.90 B:15.31 S:7.03 | R:8.26 B:12.99 S:6.51 | R:7.54 B:11.03 S:6.16 | R:4.90 B:8.87 S:4.59 |

TG: triglycerides; Non-HDL-C: non-high-density lipoprotein cholesterol; R: Roche; B: Beckman; S: Siemens.

## Concordances of the different equations for LDL-C estimation by LDL-C strata

Concordances of the different equations for LDL-C estimation by different LDL-C sublevels are given in Fig 3. For Roche direct assay, even though Friedewald and the extended Martin-Hopkins equations gave the most concordant results when LDL-C is less than 70 mg/dL, the performance of the Martin-Hopkins equation surpasses the performance of the Friedewald and the extended Martin-Hopkins equations, as well as the performances of the other equations when LDL-C level is between 70 mg/dL and 99 mg/dL. The extended Martin-Hopkins method outperformed the other equations when LDL-C level is higher than 99 mg/dL. The Martin-Hopkins method provided the most concordant results for Beckman direct assay when LDL-C measured below 70 mg/dL. However, the performance of the extended Martin-

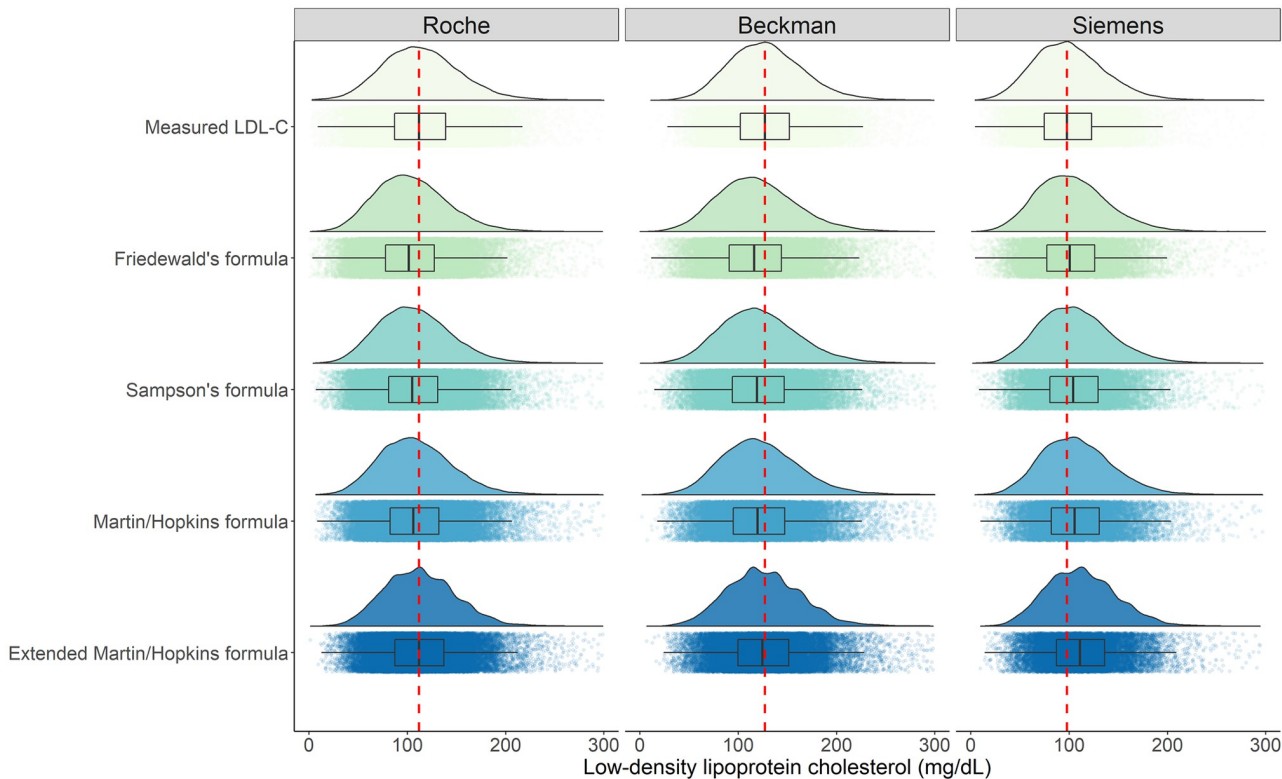

**Fig 2. Distribution density of LDL-C concentrations calculated by direct methods and different formulas.** Box-plots are also represented to compare the LDL-C levels measured by direct method with the LDL-C levels estimated by Friedewald, Sampson, Martin-Hopkins and extended Martin-Hopkins formulas. Red dash line is depicted to see the difference between direct method (Roche, Beckman or Siemens) and the formulas used to measure the LDL-C levels.

Hopkins was higher than these three methods (i.e., Friedewald, Sampson and Martin-Hopkins) when LDL-C level is above 70 mg/dL. For Siemens direct assay, the overall concordances are lower than the other assays, but extended Martin-Hopkins gave the most concordant results for LDL-C levels measured up to 129 mg/dL and Martin-Hopkins for LDL-C levels beyond 129 mg/dL.

## Concordances of the different equations for LDL-C estimation by triglycerides strata

Overall concordances for LDL-C estimates by five different triglycerides sublevels (< 100 mg/dL, 100 to 149 mg/dL, 150 to 199 mg/dL, 200 to 329 mg/dL and ≥ 400 mg/dL) were given in Fig 4. The results showed that the extended Martin-Hopkins gave the highest concordance for each assay and for any TG sublevels. However, while the TG levels increased, the concordances of the equations decreased for any direct assays. The concordance of the equations used for LDL-C estimation given five different TG sublevels with six different LDL-C strata was also calculated (S1–S6 Figs).

When the performances of the methods were evaluated according to the TG changes in LDL-C sublevels, we found that the results differed from assay to assay (S1–S6 Figs). In patients with LDL-C < 70 mg/dL, we observed that the concordances of Friedewald, Sampson and Martin-Hopkins formulas for TG < 150 mg/dL for Roche and Beckman assays were quite similar and provided the highest results compared to the extended Martin-Hopkins formula.

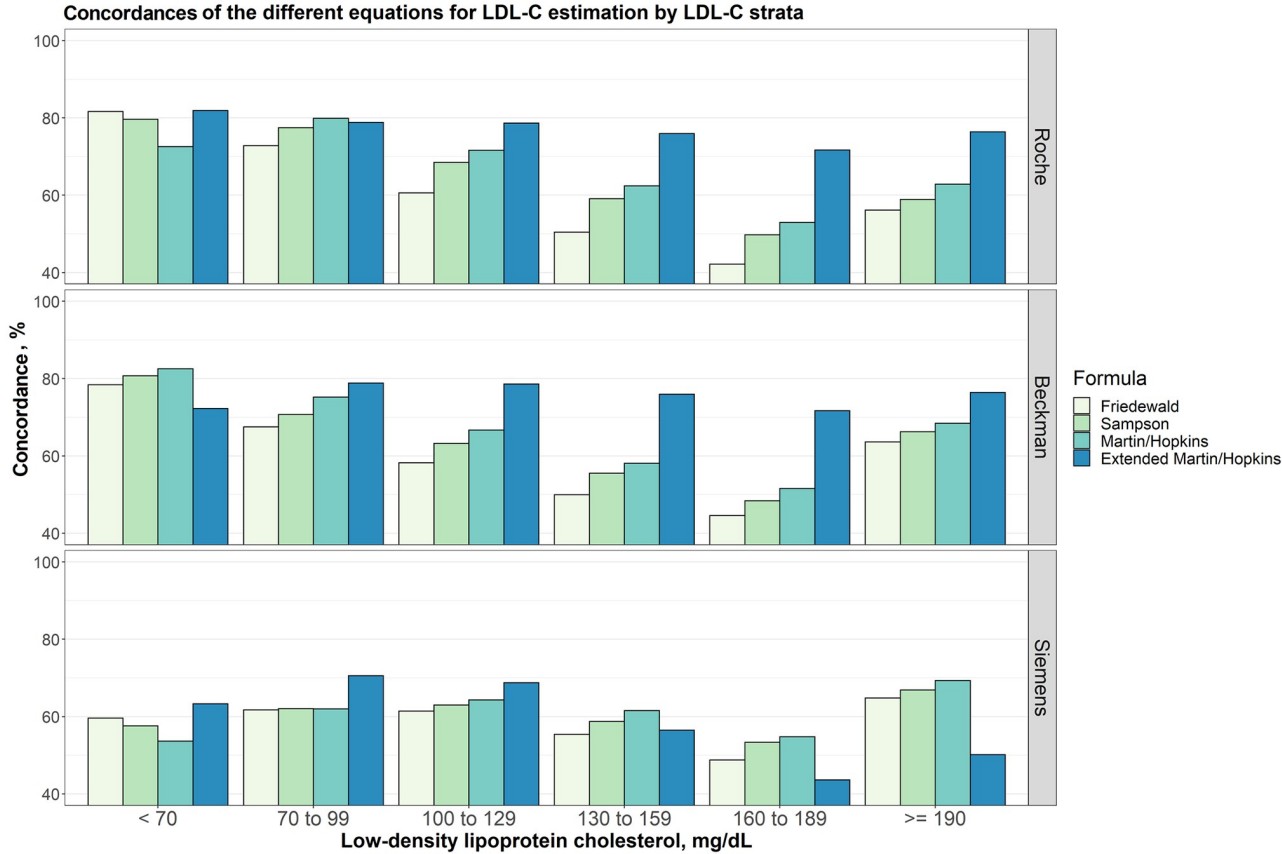

**Fig 3. Concordances of the different equations for LDL-C estimation by LDL-C strata.** Concordances of different equations for LDL-C estimation by LDL-C groups assuming different direct measures (i.e., Roche, Beckman and Siemens) are given in a clustered bar chart. Each bar indicates the concordance of estimating LDL-C levels by different formulas for each group of LDL-C levels given that the LDL-C levels measured by Roche, Beckman and Siemens direct methods, accordingly.

We found that the performance of Martin-Hopkins and Friedewald methods decreased significantly as TG increased for the Roche assay, while this decrease was comparatively less for the Sampson formula. For TG $\geq$ 400 mg/dL, Sampson equation gave the most concordant and the extended Martin-Hopkins equation gave the second most concordant results. We observed that the results of the Martin-Hopkins method were most concordant as TG increases for the Beckman assay. For this assay, we could not calculate the performances of the methods because the number of observations was very low in the LDL-C < 70 mg/dL and TG $\geq$ 400 mg/dL scenarios. For Siemens assay, while the performance of the extended Martin-Hopkins method was highest in the TG < 150 mg/dL scenario, we found a significant increase in the performance of the Friedewald method when the TG levels were between 150 and 400 mg/dL. For TG $\geq$ 400, Sampson and the extended Martin-Hopkins equations provided the best results with very similar concordances S1 Fig. For LDL-C between 70 to 99 mg/dL, in most scenario, the concordance of Martin-Hopkins equation for Roche assay and the extended Martin-Hopkins for the other assays were observed to be the highest S2 Fig. Sampson equation provided the most concordant results for Siemens assay for TG $\geq$ 400 mg/dL. It is obvious from S3 Fig that extended Martin-Hopkins method performed better than other methods for each assay with any TG sublevels when LDL-C levels were between 100 and 129 mg/dL. The same pattern can be seen for the LDL-C sublevels (130 to 159 mg/dL and 160 to 189 mg/dL) for Roche and

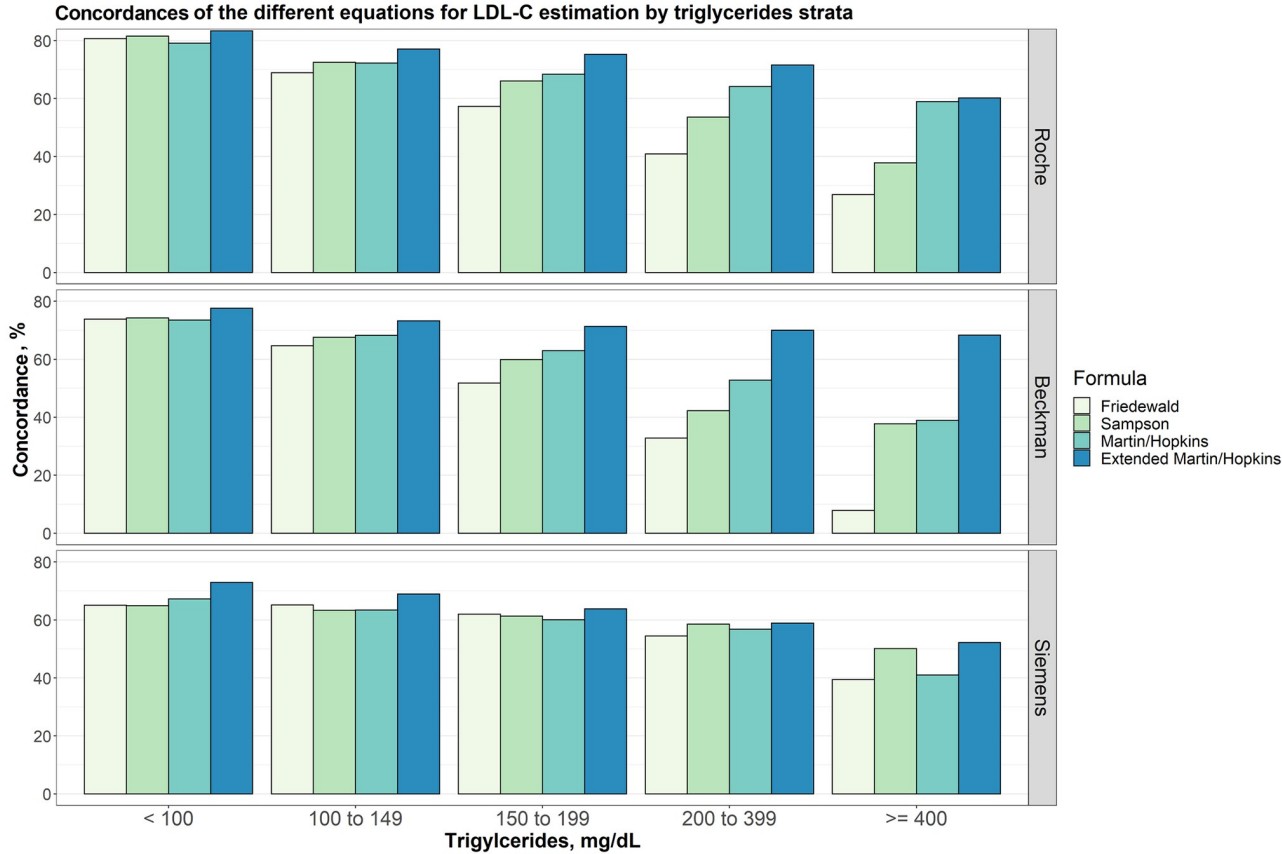

**Fig 4. Concordances of the different equations for LDL-C estimation by triglycerides strata.** Concordances of different equations for LDL-C estimation by triglycerides groups for different direct measures (i.e., Roche, Beckman and Siemens) are given in a clustered bar chart. Each bar indicates the concordance of estimating LDL-C levels by different formulas for each group of triglycerides concentration given that the LDL-C levels measured by Roche, Beckman and Siemens direct methods, accordingly.

Beckman direct assays (S4 and S5 Figs). When LDL-C is measured above 189 mg/dL by Roche direct assay, while the most concordant estimate was obtained by the Sampson method for TG < 100 mg/dL, the extended Martin Hopkins formula yielded more concordant LDL-C estimates for TG ≥ 100 mg/dL compared to other formulas S6 Fig. When LDL-C is measured above 189 mg/dL by Beckman direct assay, the Sampson formula performed better than other formulas in the first three TG sublevels whereas the extended Martin-Hopkins indicated the highest concordance of all methods applied in this study S6 Fig. When LDL-C is measured above 189 mg/dL by Siemens direct assay, the extended Martin-Hopkins method was the least concordant method with the lowest concordance for any TG sublevels S6 Fig.

## Concordance of the different equations for LDL-C estimation in patients with low LDL-C and/or higher TG levels

There are few studies on the validity of LDL-C estimation equations for patients with low LDL-C (i.e., LDL-C < 70 mg/dL) and/or higher TG levels (i.e., TG ≥ 400 mg/dL). For this reason, a special attention is given to the comparison of equations for LDL-C < 70 mg/dL and/or TG ≥ 400 mg/dL. As seen from Table 3 the extended Martin-Hopkins formula for the Roche and Siemens assays, and Martin-Hopkins formula for Beckman assay gave the most concordant results for patients with low LDL-C levels (Table 3, Fig 3). For patients with higher TG

**Table 3. The concordances of LDL-C estimation equations in patients with low LDL-C and/or higher TG levels.**

| Patient group / Direct assay | Friedewald | Sampson | Martin-Hopkins | Extended Martin-Hopkins |
|---|---|---|---|---|
| **LDL-C levels < 70 mg/dL** | | | | |
| **Roche** | 0.817 | 0.797 | 0.726 | **0.819** |
| **Beckman** | 0.784 | 0.807 | **0.825** | 0.723 |
| **Siemens** | 0.596 | 0.576 | 0.537 | **0.633** |
| **TG levels ≥ 400 mg/dL** | | | | |
| **Roche** | 0.269 | 0.378 | 0.589 | **0.602** |
| **Beckman** | 0.078 | 0.378 | 0.389 | **0.683** |
| **Siemens** | 0.394 | 0.501 | 0.410 | **0.522** |
| **20cmLDL-C levels < 70 mg/dL and TG levels ≥ 400 mg/dL** | | | | |
| **Roche** | 0.318 | **0.588** | 0.306 | 0.400 |
| **Beckman** | NC | NC | NC | NC |
| **Siemens** | 0.309 | **0.545** | 0.164 | **0.545** |

NC: Not computed due to the very low sample size. The most concordant results are shown in bold.

levels, the extended Martin-Hopkins formula provided the most concordant results (Table 3, Fig 4). When the results were examined for patients with low LDL-C levels and higher TG levels, Sampson formula provided the most concordant results for Roche assay. For Siemens assay, both Sampson and the extended Martin-Hopkins formula gave the same and most concordant results (Table 3, S1 Fig).

## Concordances of the different equations for LDL-C estimation by non-HDL-C strata

Overall concordances for LDL-C estimates by six different non-HDL-C sublevels (< 100 mg/dL, 100 to 129 mg/dL, 130 to 159 mg/dL, 160 to 189 mg/dL, 190 to 219 mg/dL and ≥ 220 mg/dL) were given in Fig 5. The results showed that extended Martin-Hopkins gave the highest concordance for each assay and for almost all non-HDL-C sublevels. Additionally, for Siemens direct assay, while the non-HDL-C levels increased, the concordance of the methods decreased. The concordance of the methods used for LDL-C estimation given six different non-HDL-C sublevels with six different LDL-C strata was also evaluated (S7–S12 Figs).

## Regression analysis between LDL-C levels estimated by formulas and directly measured LDL-C levels

The linear regression analyses demonstrated that estimated LDL-C levels by extended Martin-Hopkins formula indicate better correlation with Roche and Beckman assays with an R square of 0.91 (Fig 6). The highest R square statistic was obtained with the Martin-Hopkins and the extended Martin-Hopkins methods for Siemens direct measurement method. It is obvious to see that Martin-Hopkins and extended Martin-Hopkins show a better association with any direct methods overall.

## Residual error plots for LDL-C by different formulas with respect to different direct assay methods

The residual error plots show how the bias between LDL-C estimations calculated by the equations and direct measurements varies according to triglyceride levels (Fig 7). It is seen that the Friedewald formula underestimated the LDL-C levels as the TG level increased in all assays.

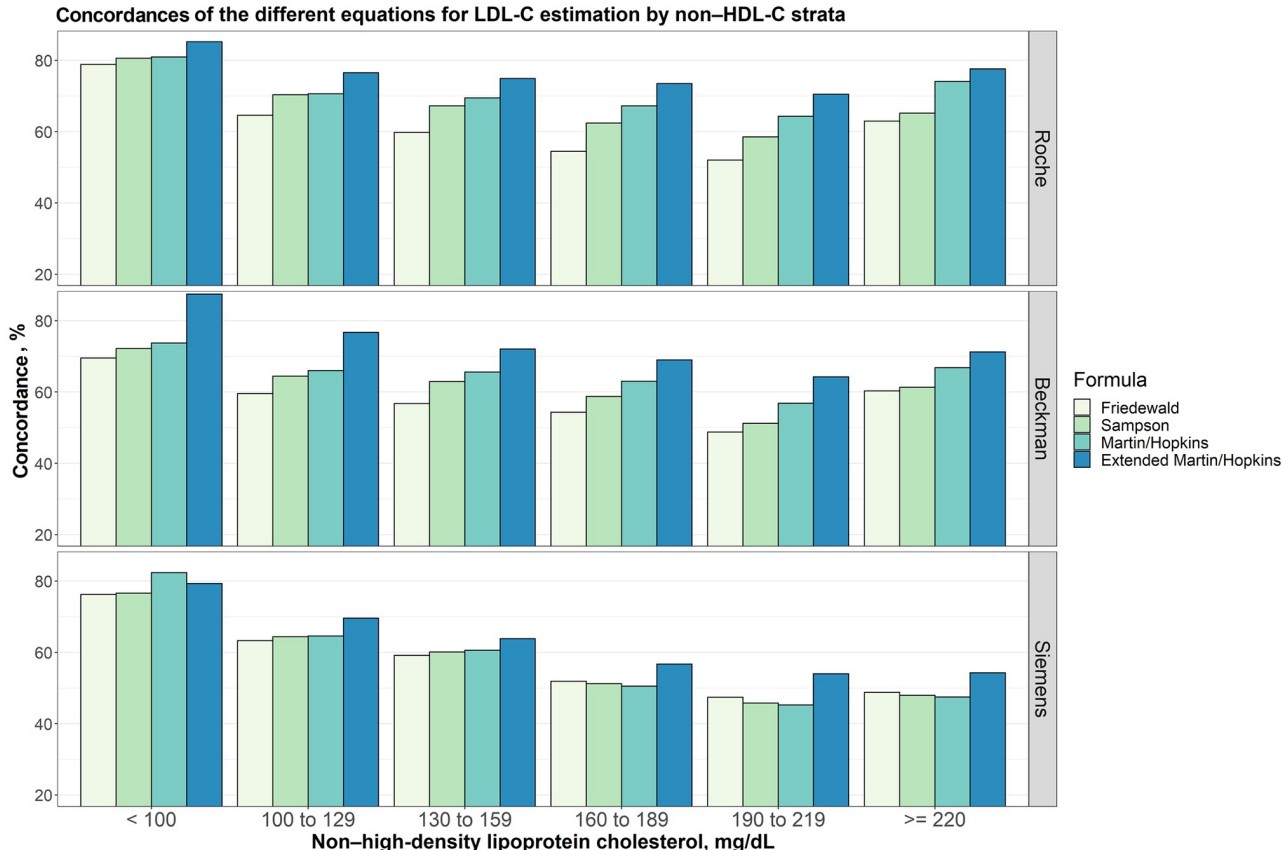

**Fig 5. Concordances of the different equations for LDL-C estimation by non-HDL-C strata.** Concordances of different equations for LDL-C estimation by non-HDL-C groups assuming different direct measures (i.e., Roche, Beckman and Siemens) are given in a clustered bar chart. Each bar indicates the concordance of estimating LDL-C levels by different formulas for each group of non-HDL-C concentration given that the LDL-C levels measured by Roche, Beckman and Siemens direct methods, accordingly.

Similar results were found in the Sampson formula for the Roche and Beckman assays, and in the Martin-Hopkins formula for the Beckman assay. It was observed that the bias in the Siemens assay according to the TG levels was less compared to the other assays. In the Roche and Siemens formulas, the bias for the Martin-Hopkins formula was found to be constant according to the change in TG levels. For the Extended Martin-Hopkins formula, it was determined that the bias in all assays did not change according to TG levels and was close to zero. The lowest mean absolute deviation statistics were obtained with the extended Martin-Hopkins formula for all assays.

## Discussion

The Martin-Hopkins formula has been proposed as a replacement for the Friedewald formula and its validity has been proven in many populations. However, it has been reported that this formula is inaccurate in patients with hypertriglyceridemia [18]. Today, the most recent formula for estimating LDL-C level has been proposed by Sampson et al. [18]. Recent studies have investigated the validity of the Sampson formula in different populations and evaluated whether it produces more accurate results compared to the Martin-Hopkins formula. Song et al. showed that the Martin-Hopkins method gave better results in the East Asian population [20]. Most of the studies stated that the Sampson formula gave the most promising results

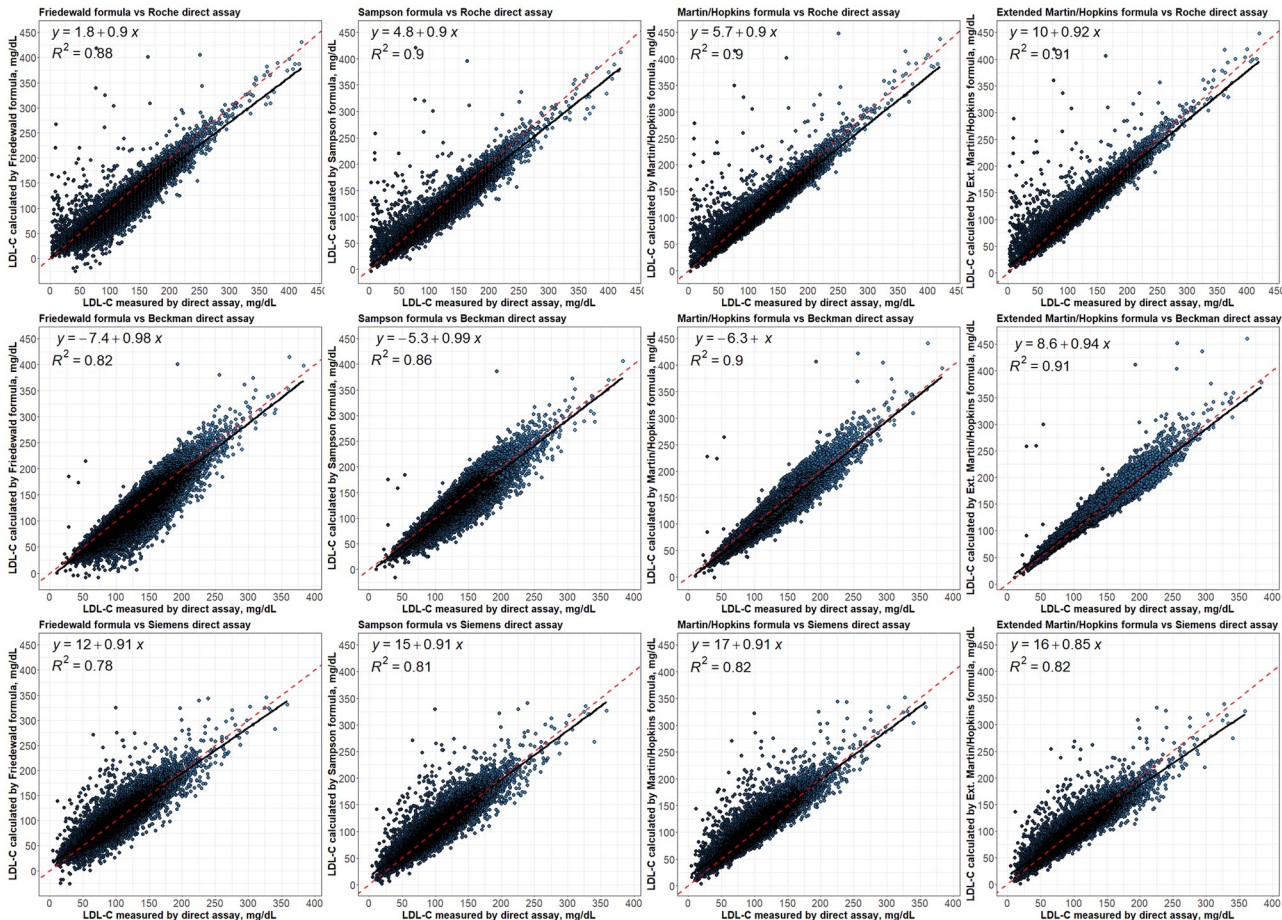

**Fig 6. Regression analysis between LDL-C levels estimated by formulas and directly measured LDL-C levels.** Correlations of estimated LDL-C levels by Friedewald, Sampson, Martin-Hopkins and extended Martin-Hopkins formulas with LDL-C levels directly measured by Roche, Beckman and Siemens.

compared to other formulas [21–24]. In one study, it was stated that both Sampson and Martin-Hopkins methods provide the most accurate results [25].

Our study contributes to the literature by validating and comparing Sampson's formula with the Friedewald and Martin-Hopkins equations in different direct assays for estimation of LDL-C. When we evaluated the general concordance of the methods with different direct assays in our study, we observed that the Martin-Hopkins method gave slightly more concordant results for the Roche and Siemens direct assays, while the methods produced similar results for the Beckman direct assay. However, when we used the median statistics obtained from our own population, we observed that the extended Martin-Hopkins approach produced much more concordant results in all assays compared to the Friedewald, Martin-Hopkins and Sampson methods. Even with the median coefficients obtained from the 30 and 70-cell tables, we observed a significant increase in concordance compared to the other methods. We observed that the concordances were very similar in tables with cells 70, 130, 180 and 420 where the median statistics were calculated.

Although we determined that the extended Martin-Hopkins method was the most concordant method for all assays, as a new finding, we found that the results were variable in different assays. For the Beckman assay, the extended Martin-Hopkins method gave the lowest concordance at

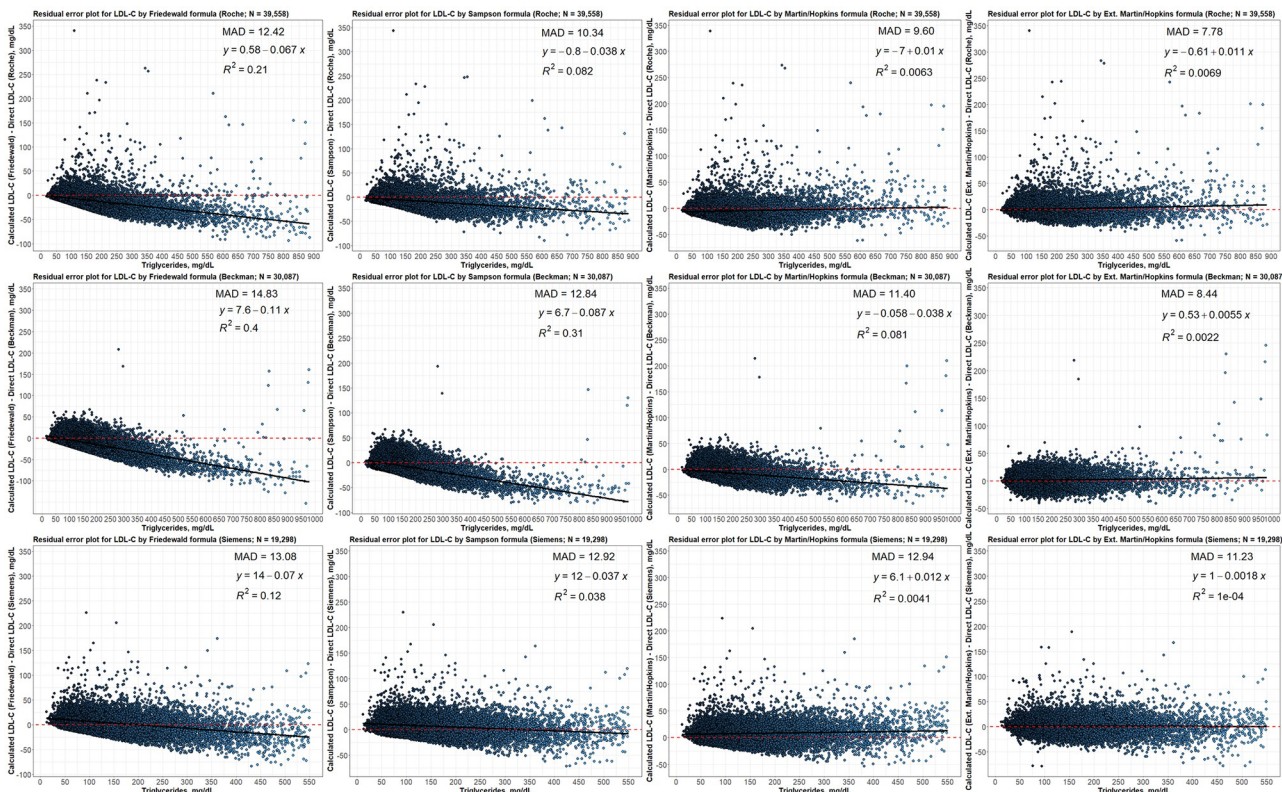

**Fig 7. Residual error plots for LDL-C by different formulas concerning to different direct assay methods.** While the values on x-axis show TG levels, the values on y-axis shows the difference between estimated LDL-C (by Friedewald, Sampson, Martin-Hopkins or extended Martin-Hopkins) and direct LDL-C levels (calculated by Roche, Beckman or Siemens). The mean absolute deviation (MAD) for each possible case is also given in each panel for each dataset.

low values of LDL-C, while it became the most concordant as the LDL-C level increased. The most concordant results were obtained with the extended Martin-Hopkins method in many cases for the Roche assay. For the Siemens assay, the results were vice versa. In this assay, the performance of the extended Martin-Hopkins method was found to be the best when the LDL-C level was < 130 mg/dL, and the lowest when the LDL-C level was > 130 mg/dL. This low performance may be sourced from the low sample size in these scenarios. LDL-C level was between 130–159 mg/dL in 13.4% of individuals directly measured with the Siemens assay, between 160–189 mg/dL in 4.4% of individuals, and 190 mg/dL and above in 1.5% of individuals. We observed that the extended Martin-Hopkins method still performed best, although there was variation in the performance of the methods as the triglyceride levels and non-HDL-C levels changed. In a previous study made by Rossouw et al. [26], the comparability of direct LDL-C levels obtained Abbott Architect and the Roche Cobas analyzers with the Martin Hopkins, Sampson and Friedewald equations were evaluated, and they found that the performance of predictive equations could be influenced according to the platform and LDL-C levels. Besides, they concluded that Martin-Hopkins formula can be safely implemented for both Abbott and Roche platforms. These findings are in accordance with our findings.

Sampson reported that their equation is more accurate than Martin-Hopkins equation for patients with a low LDL-C level and/or hypertriglyceridemia. In our study, when the LDL-C level is below 70 mg/dL, the concordance of the Martin-Hopkins method for Beckman assay and the extended Martin-Hopkins method for Roche and Siemens assays were found to be the

highest. The concordance of the extended Martin-Hopkins method is highest for all assays in individuals with triglyceride levels above 400 mg/dL. The concordance of the Sampson method for Roche assay was found to be highest in patients with LDL-C lower than 70 mg/dL and triglyceride level higher than 400 mg/dL. In the same scenario, the concordance of Sampson and extended Martin-Hopkins methods for Siemens assay were found to be similar and the highest. Therefore, as Sampson et al. [18] stated, our findings for the Roche assay support the statement that the Sampson formula performs best in patients with "low LDL-C level and hypertriglyceridemia (TG levels ≥ 400 mg/dL)". However, we do not agree with the statement of Sampson et al. [18] that the Sampson formula performs best in patients with "low LDL-C level or hypertriglyceridemia (TG levels ≥ 400 mg/dL)". In this scenario, the extended Martin-Hopkins approach, calculated from the median statistics of our population, gives the best results in all platforms.

The accurate estimation of LDL-C is important to identify the patients who have risk for future cardiovascular disease and the success of the treatment in patients who take lipid lowering drugs [27]. We determined the differences between the measured and estimated LDL-C levels on three different platforms. The direction and magnitude of the errors varied with the different analytical platforms and LDL-C strata. Accordingly, laboratories should determine the best formula in LDL-C estimation for their analytical platform.

In the present study, there was a difference in terms of the concordances between formula in patients with < 70 mg/dL LDL-C levels. Accurate estimation of LDL-C levels < 70 mg/dL is becoming increasingly important especially in patients at risk of developing cardiovascular disease (and with the increasing use of PCSK-inhibitors). Adult Treatment Panel (ATP III) guidelines recommend using a LDL-C threshold of 70 mg/dL in patients with very high-risk atherosclerotic cardiovascular disease and multiple risk conditions. To achieve LDL-C levels below 70 mg/dL and absolute risk reduction combining a statin with PCSK9 inhibitor therapy are being recommended [16]. In our study, the extended Martin-Hopkins formula more closely estimated LDL-C levels for samples with LDL-C lower than 70 mg/dL in Roche and Siemens platforms as compared to Friedewald, Martin-Hopkins and Sampson formula. In Beckman platform, the Martin-Hopkins formula more closely estimated LDL-C levels for samples with LDL-C lower than 70 mg/dL. These data suggest that the extended Martin-Hopkins formula for Roche and Siemens platforms, and Martin-Hopkins formula for Beckman platforms may be preferred to estimate LDL-C levels in patients with very high-risk atherosclerotic cardiovascular disease and multiple risk conditions.

In this study, we did not use beta quantification or preparative ultracentrifugation methods in the determination of the LDL-C levels. Since these techniques are expensive and requiring highly manual technique, direct assays are the most used methods in the determination of LDL-C. However, lack of standardization is important problems for these assays. Therefore, our hypothesis regarding with the calculation of the median statistics from their own population should be evaluated with this limitation.

## Conclusion

Although the results of the methods in different assays may vary, the extended Martin-Hopkins approach is the method with the highest overall concordance. When using the Martin-Hopkins formula, calculation of the median statistics from their own population might be helpful to researchers to obtain more concordant results with the direct assays. Also, the extended Martin-Hopkins approach is the best approach for patients with a low LDL-C level or higher TG levels. Further validation is warranted in different populations.

## Supporting information

**S1 Fig. Concordances of the different equations for LDL-C estimation by triglycerides strata (LDL $<$ 70 mg/dL).**
(PNG)

**S2 Fig. Concordances of the different equations for LDL-C estimation by triglycerides strata (LDL between 70 to 99 mg/dL).**
(PNG)

**S3 Fig. Concordances of the different equations for LDL-C estimation by triglycerides strata (LDL between 100 to 129 mg/dL).**
(PNG)

**S4 Fig. Concordances of the different equations for LDL-C estimation by triglycerides strata (LDL between 130 to 159 mg/dL).**
(PNG)

**S5 Fig. Concordances of the different equations for LDL-C estimation by triglycerides strata (LDL between 160 to 189 mg/dL).**
(PNG)

**S6 Fig. Concordances of the different equations for LDL-C estimation by triglycerides strata (LDL $\geq$ 190 mg/dL).**
(PNG)

**S7 Fig. Concordances of the different equations for LDL-C estimation by nonHDL-C strata (LDL $<$ 70 mg/dL).**
(PNG)

**S8 Fig. Concordances of the different equations for LDL-C estimation by nonHDL-C strata (LDL between 70 to 99 mg/dL).**
(PNG)

**S9 Fig. Concordances of the different equations for LDL-C estimation by nonHDL-C strata (LDL between 100 to 129 mg/dL).**
(PNG)

**S10 Fig. Concordances of the different equations for LDL-C estimation by nonHDL-C strata (LDL between 130 to 159 mg/dL).**
(PNG)

**S11 Fig. Concordances of the different equations for LDL-C estimation by nonHDL-C strata (LDL between 160 to 189 mg/dL).**
(PNG)

**S12 Fig. Concordances of the different equations for LDL-C estimation by nonHDL-C strata (LDL $\geq$ 190 mg/dL).**
(PNG)

**S1 File. Median statistics for the ratio of triglycerides to very low-density lipoprotein cholesterol by the cross table of non-high-density lipoprotein cholesterol and triglycerides calculated from the Turkish population.**
(DOCX)

## Author Contributions

**Conceptualization:** Gözde Ertürk Zararsız, Halef Okan Doğan, Gökmen Zararsız.

**Data curation:** Gözde Ertürk Zararsız, Serkan Bolat, Ahu Cephe, Serra İlayda Yerlitaş.

**Formal analysis:** Gözde Ertürk Zararsız, Ahu Cephe, Serra İlayda Yerlitaş.

**Funding acquisition:** Gökmen Zararsız.

**Investigation:** Gözde Ertürk Zararsız.

**Methodology:** Gözde Ertürk Zararsız, Ahu Cephe, Necla Kochan, Halef Okan Doğan.

**Project administration:** Halef Okan Doğan, Gökmen Zararsız.

**Resources:** Serkan Bolat.

**Software:** Gözde Ertürk Zararsız.

**Supervision:** Gökmen Zararsız.

**Validation:** Gözde Ertürk Zararsız, Serkan Bolat, Necla Kochan, Halef Okan Doğan.

**Visualization:** Gözde Ertürk Zararsız, Ahu Cephe, Serra İlayda Yerlitaş.

**Writing – original draft:** Gözde Ertürk Zararsız, Serkan Bolat, Necla Kochan, Halef Okan Doğan, Gökmen Zararsız.

**Writing – review & editing:** Halef Okan Doğan, Gökmen Zararsız.

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
