## [Decision Letter · Decision Letter 0]

1 Mar 2022

PONE-D-22-02690Validation of Friedewald, Martin/Hopkins and Sampson Equations in the Low-Density Lipoprotein Cholesterol Estimation with Different AssaysPLOS ONE

Dear Dr. Zararsiz,

Thank you for submitting your manuscript to PLOS ONE. After careful consideration, we feel that it has merit but does not fully meet PLOS ONE’s publication criteria as it currently stands. Therefore, we invite you to submit a revised version of the manuscript that addresses the points raised during the review process.

Your manuscript is very interesting and thus altogether 4 refvviewers gave comments.  They have minor comments or suggestions to improve your manuscript.  Please check them and reply accordingly and clearly mark you changes.

We look forward to receiving your revised manuscript.

Kind regards,

Katriina Aalto-Setala, Professor

Academic Editor

PLOS ONE

Journal Requirements:

Reviewers' comments:

Reviewer's Responses to Questions

**Comments to the Author**

1. Is the manuscript technically sound, and do the data support the conclusions?

Reviewer #1: Yes

Reviewer #2: Yes

Reviewer #3: Yes

Reviewer #4: Yes

2. Has the statistical analysis been performed appropriately and rigorously? 

Reviewer #1: Yes

Reviewer #2: Yes

Reviewer #3: Yes

Reviewer #4: Yes

3. Have the authors made all data underlying the findings in their manuscript fully available?

Reviewer #1: Yes

Reviewer #2: Yes

Reviewer #3: Yes

Reviewer #4: No

4. Is the manuscript presented in an intelligible fashion and written in standard English?

Reviewer #1: Yes

Reviewer #2: Yes

Reviewer #3: Yes

Reviewer #4: Yes

5. Review Comments to the Author

Reviewer #1: The article evaluates the performance and compares the different formulas available for the calculation of LDL cholesterol in a population of 88,943 participants, which allows inferring the advantages of the various formulas in general and according to special situations such as hypertriglyceridemia. I would like to see the applicability of these formulas in a subgroup of patients with Diabetes Mellitus.

From my review, the article can be accepted for publication.

Reviewer #2: Comprehensive retrospective validation study of large dataset of LDL-C calculated with novel formulas compared to direct LDL-C assays most widely available in the clinical laboratories.

Like direct LDL-C assays, the direct HDL-C measurements (used in the LDL-C formulas) vary between methods and manufacters used and contribute to uncertaintly of calculated LDL-C, especially in hypertriglyceridemic samples. The methods section should clarify which HDL-C assays were used in the LDL-C calculations for each comparison with direct LDL-C: presumably the same manufacturer (Roche, Siemens, Beckman) for both HDL-C and LDL-C assays was used in each comparison of calculated vs. direct LDL-C?

The conclusion about accuracy is unwarranted, since LDL-C data were not compared vs. the reference method (beta-quantification or ultracentrifugation-based LDL-C). None of the direct LDL-C assays used in this study (Roche, Siemens, Beckman) can be considered to represent "true" LDL-C (beta-quantification). This study can only provide evidence of concordance of LDL-C formulas with direct LDL-C.

Reviewer #3: Zararsız et al analyzed a total of 88,943 adults at Sivas Cumhuriyet University Hospital (Sivas, Turkey) for their LDL-C levels by various equations vs various direct homogeneous assays. The accuracies between the estimations and direct measurements were evaluated overall and separately for the LDL-C, TG and non-HDL-C sublevels. Linear regression analysis was applied and residual error plots were generated between each estimation and direct measurement method. Coefficient of determination (R2) and mean absolute deviations were also examined. In total, the results support the extended Martin/Hopkins approach as having the highest overall accuracy. These results are important because LDL-C is central in clinical management.

Suggestions:

Simplify the title to: Validation of Friedewald, Martin/Hopkins and Sampson Low-Density Lipoprotein Cholesterol Equations

In the abstract, recommend deleting “Although the best performance was obtained by the Sampson formula in patients with ”low LDL-C level and hypertriglyceridemia” since this is confusing and the data are not strong in support of this.

The Abstract conclusion is confusingly written whereas the main manuscript text is more clear. The main text reads, “Although the results of the methods in different assays may vary, the extended Martin/Hopkins approach is the method with the highest overall accuracy.” Recommend using a similar sentence in the conclusion of the Abstract as it concisely captures to two main findings: overall best performance of extended Martin/Hopkins and variability between direct assays.

Reference 16 is an important study and currently not covered sufficiently in the paper: Martin SS, Giugliano RP, Murphy SA, al. Comparison of Low-Density Lipoprotein Cholesterol Assessment by Martin/Hopkins Estimation, Friedewald Estimation, and Preparative Ultracentrifugation: Insights From the FOURIER Trial. JAMA Cardiology. 2018;3(8):749–753. doi:10.1001/jamacardio.2018.1533.

This study shows in a large-scale clinical trial population with low LDL-C and high TGs, the high performance of the Martin/Hopkins compared with preparative ultracentrifugation. This is some of the best evidence for the Martin/Hopkins equation in intensively treated high risk patients. Furthermore, it is solid data on low LDL-C level and hypertriglyceridemia (150-399 mg/dL).

Furthermore, the manuscript seems to use the term “hypertriglyceridemia” to mean TG levels of 400 mg/dL or greater. However, this is really a hypertriglyceridemia of severe magnitude. TG levels of 150 mg/dL are considered hypertriglyceridemia in clinical practice, just of lesser magnitude. Recommend more clear use of the term along these lines and not restricting its use to TG levels of 400 mg/dL or greater.

In the Introduction, it currently reads, “In 2020, in order to accurately estimate LDL-C levels for patients with TG levels up to 800 mg/dL, Sampson et al. derived a new novel equation (hereafter referred to as the Sampson formula) which uses β-quantification results obtained out of a population with a high frequency of hypertriglyceridemia [18].” – recommend changing “in order to accurately estimate LDL-C levels” to “with an aim of estimating LDL-C levels”

Recommend softening this sentence: “When using the Martin/Hopkins formula, researchers must calculate the median statistics from their own population.” – Although this seems attractive, the challenge is that the difference between the original study and one's own population might not be due to the population, but rather the use of less reliable direct homogeneous assays in one's own population vs ultracentrifugation in the original Martin/Hopkins derivation. There should be more caution advised in the paper about direct homogeneous assays. As the introduction notes, "clinical laboratories perform direct chemical assays in order to measure the LDL-C levels at higher TG levels, but these

direct assays lack standardization, are time consuming and costly." This lack of standardization is clearly seen in the results of this paper and warrants caution, rather than assuming that median statistics are best re-calculated.

Recommend deleting this sentence from the conclusion as there are insufficient data to support it: “In patients with a low LDL-C level and hypertriglyceridemia, the accuracy of the Sampson method was found to be the best method for the Roche assay.”

Reviewer #4: The manuscript is a good piece of work. The authors have included a massive amount of data. I do, however, do not think that they should use the word "accuracy" since we do not have a true target value. They should consider focussing on the term "comparability" between the calculation and an enzymatic assay.

I think they should also elaborate on why accurate LDL-C estimation in the <70mg/dL is important eg. PCSK-9 inhibitors and the lower LDL-C targets being recommended by the Fourier trial etc.

And they can refer to the article by Rossouw et al in Clinical Chemistry and Laboratory Medicine with regards to Martin-Hopkins performance being more consistent across different analytical platforms.

Please refer to the additional comments on the manuscript PDF.

6. PLOS authors have the option to publish the peer review history of their article (what does this mean?). If published, this will include your full peer review and any attached files.

Reviewer #1: No

Reviewer #2: No

Reviewer #3: No

Reviewer #4: No

---

## [Author Response · Author response to Decision Letter 0]

2 Apr 2022

Response to Reviewers

Reviewer #1: The article evaluates the performance and compares the different formulas available for the calculation of LDL cholesterol in a population of 88,943 participants, which allows inferring the advantages of the various formulas in general and according to special situations such as hypertriglyceridemia. I would like to see the applicability of these formulas in a subgroup of patients with Diabetes Mellitus.

From my review, the article can be accepted for publication.

Response: We thank the referee for these valuable comments. Investigating the validity of formulas in Diabetes Mellitus patients is an issue that we also have in mind and that we want to evaluate in future studies. We have designed a study related to this topic. We will be happy if the reviewer contributes to this study as well.

##

Reviewer #2: Comprehensive retrospective validation study of large dataset of LDL-C calculated with novel formulas compared to direct LDL-C assays most widely available in the clinical laboratories.

Like direct LDL-C assays, the direct HDL-C measurements (used in the LDL-C formulas) vary between methods and manufacters used and contribute to uncertaintly of calculated LDL-C, especially in hypertriglyceridemic samples. The methods section should clarify which HDL-C assays were used in the LDL-C calculations for each comparison with direct LDL-C: presumably the same manufacturer (Roche, Siemens, Beckman) for both HDL-C and LDL-C assays was used in each comparison of calculated vs. direct LDL-C?

Response: HDL-C assays were also measured with the same manufacturers. The information related with the analytical methods of HDL-C measurement were given in the Methods section as follows: 

Lipid measurements 

Different systems were used to directly measure HDL-C, LDL-C, triglycerides, and total cholesterol parameters. Detailed measurement procedures were given below according to used systems. 

Roche Cobas 8000, c-702 and c-501: Total cholesterol, triglycerides and LDL-C, HDL-C measurements were performed using colorimetric enzymatic reaction. 

Siemens Advia 1800: HDL-C levels were determined with Trinder reaction. Triglycerides and LDL-C, total cholesterol measurements were performed using colorimetric enzymatic reaction.

Beckman Coulter AU5800: Total cholesterol, triglycerides and LDL-C, HDL-C measurements were performed using colorimetric enzymatic reaction.

The conclusion about accuracy is unwarranted, since LDL-C data were not compared vs. the reference method (beta-quantification or ultracentrifugation-based LDL-C). None of the direct LDL-C assays used in this study (Roche, Siemens, Beckman) can be considered to represent "true" LDL-C (beta-quantification). This study can only provide evidence of concordance of LDL-C formulas with direct LDL-C.

Response: We thank the referee for this valuable comment. In Discussion section, we added this issue as a limitation of the study. We have also changed the term of ‘accuracy’ with ‘concordance’ term.

##

Reviewer #3: Zararsız et al analyzed a total of 88,943 adults at Sivas Cumhuriyet University Hospital (Sivas, Turkey) for their LDL-C levels by various equations vs various direct homogeneous assays. The accuracies between the estimations and direct measurements were evaluated overall and separately for the LDL-C, TG and non-HDL-C sublevels. Linear regression analysis was applied and residual error plots were generated between each estimation and direct measurement method. Coefficient of determination (R2) and mean absolute deviations were also examined. In total, the results support the extended Martin/Hopkins approach as having the highest overall accuracy. These results are important because LDL-C is central in clinical management.

Suggestions:

Simplify the title to: Validation of Friedewald, Martin/Hopkins and Sampson Low-Density Lipoprotein Cholesterol Equations

Response: The study title is simplified as suggested.

In the abstract, recommend deleting “Although the best performance was obtained by the Sampson formula in patients with ”low LDL-C level and hypertriglyceridemia” since this is confusing and the data are not strong in support of this.

Response: This sentence is deleted as suggested by the reviewer.

The Abstract conclusion is confusingly written whereas the main manuscript text is more clear. The main text reads, “Although the results of the methods in different assays may vary, the extended Martin/Hopkins approach is the method with the highest overall accuracy.” Recommend using a similar sentence in the conclusion of the Abstract as it concisely captures to two main findings: overall best performance of extended Martin/Hopkins and variability between direct assays.

Response: The conclusion part of the Abstract section is revised as suggested.

Reference 16 is an important study and currently not covered sufficiently in the paper: Martin SS, Giugliano RP, Murphy SA, al. Comparison of Low-Density Lipoprotein Cholesterol Assessment by Martin/Hopkins Estimation, Friedewald Estimation, and Preparative Ultracentrifugation: Insights From the FOURIER Trial. JAMA Cardiology. 2018;3(8):749–753. doi:10.1001/jamacardio.2018.1533.

We inserted new paragraph by using this article. Related paragraph was given below;

Adult Treatment Panel (ATP III) guidelines recommend using a LDL-C threshold of 70 mg/dL in patients with very high-risk atherosclerotic cardiovascular disease and multiple risk conditions. To achieve LDL-C levels below 70 mg/dL and absolute risk reduction combining a statin with PCSK9 inhibitor therapy are being recommended (doi:10.1001/jamacardio.2018.1533). In our study, the extended Martin/Hopkins formula more closely estimated LDL-C levels for samples with LDL-C lower than 70 mg/dL in Roche and Siemens platforms as compared to Friedewald, Martin/Hopkins and Sampson formula. In Beckman platform, the Martin/Hopkins formula more closely estimated LDL-C levels for samples with LDL-C lower than 70 mg/dL. These data suggest that the extended Martin/Hopkins formula for Roche and Siemens platforms, and Martin/Hopkins formula for Beckman platforms may be preferred to estimate LDL-C levels in patients with very high-risk atherosclerotic cardiovascular disease and multiple risk conditions.

This study shows in a large-scale clinical trial population with low LDL-C and high TGs, the high performance of the Martin/Hopkins compared with preparative ultracentrifugation. This is some of the best evidence for the Martin/Hopkins equation in intensively treated high risk patients. Furthermore, it is solid data on low LDL-C level and hypertriglyceridemia (150-399 mg/dL).

Response: We thank to the reviewer for his/her valuable comment.

Furthermore, the manuscript seems to use the term “hypertriglyceridemia” to mean TG levels of 400 mg/dL or greater. However, this is really a hypertriglyceridemia of severe magnitude. TG levels of 150 mg/dL are considered hypertriglyceridemia in clinical practice, just of lesser magnitude. Recommend more clear use of the term along these lines and not restricting its use to TG levels of 400 mg/dL or greater.

Response:

We clarified this issue on the manuscript as suggested.

In the Introduction, it currently reads, “In 2020, in order to accurately estimate LDL-C levels for patients with TG levels up to 800 mg/dL, Sampson et al. derived a new novel equation (hereafter referred to as the Sampson formula) which uses β-quantification results obtained out of a population with a high frequency of hypertriglyceridemia [18].” – recommend changing “in order to accurately estimate LDL-C levels” to “with an aim of estimating LDL-C levels”

Response: This sentence is revised as suggested.

Recommend softening this sentence: “When using the Martin/Hopkins formula, researchers must calculate the median statistics from their own population.” – Although this seems attractive, the challenge is that the difference between the original study and one's own population might not be due to the population, but rather the use of less reliable direct homogeneous assays in one's own population vs ultracentrifugation in the original Martin/Hopkins derivation. There should be more caution advised in the paper about direct homogeneous assays. As the introduction notes, "clinical laboratories perform direct chemical assays in order to measure the LDL-C levels at higher TG levels, but these direct assays lack standardization, are time consuming and costly." This lack of standardization is clearly seen in the results of this paper and warrants caution, rather than assuming that median statistics are best re-calculated.

Since we did not use the gold standard approaches, we added a limitation paragraph on this issue. We softened the sentence as recommended by the reviewer. 

Discussion:

In this study we did not use beta quantification or preparative ultracentrifugation methods in the determination of the LDL-C levels. Since these techniques are expensive and requiring highly manual technique, direct assays are the most used methods in the determination of LDL-C. However, lack of standardization is important problems for these assays. Therefore, our hypothesis regarding with the calculation of the median statistics from their own population should be evaluated with this limitation. 

Conclusion:

When using the Martin/Hopkins formula, calculation of the median statistics from their own population might be helpful to researchers to obtain more concordant results with the direct assays.

Recommend deleting this sentence from the conclusion as there are insufficient data to support it: “In patients with a low LDL-C level and hypertriglyceridemia, the accuracy of the Sampson method was found to be the best method for the Roche assay.”

We deleted this sentence as suggested.

##

Reviewer #4: The manuscript is a good piece of work. The authors have included a massive amount of data. I do, however, do not think that they should use the word "accuracy" since we do not have a true target value. They should consider focussing on the term "comparability" between the calculation and an enzymatic assay.

Response: We thank the referee for this valuable comment. In Discussion section, we added this issue as a limitation of the study. We changed the term of ‘accuracy’ with ‘concordance’ term.

I think they should also elaborate on why accurate LDL-C estimation in the <70mg/dL is important eg. PCSK-9 inhibitors and the lower LDL-C targets being recommended by the Fourier trial etc.

We inserted a paragraph related with the importance of the LDL-C estimation in the < 70mg/dL threshold.

Adult Treatment Panel (ATP III) guidelines recommend using a LDL-C threshold of 70 mg/dL in patients with very high-risk atherosclerotic cardiovascular disease and multiple risk conditions. To achieve LDL-C levels below 70 mg/dL and absolute risk reduction combining a statin with PCSK9 inhibitor therapy are being recommended (doi:10.1001/jamacardio.2018.1533). In our study, the extended Martin/Hopkins formula more closely estimated LDL-C levels for samples with LDL-C lower than 70 mg/dL in Roche and Siemens platforms as compared to Friedewald, Martin/Hopkins and Sampson formula. In Beckman platform, the Martin/Hopkins formula more closely estimated LDL-C levels for samples with LDL-C lower than 70 mg/dL. These data suggest that the extended Martin/Hopkins formula for Roche and Siemens platforms, and Martin/Hopkins formula for Beckman platforms may be preferred to estimate LDL-C levels in patients with very high-risk atherosclerotic cardiovascular disease and multiple risk conditions.

And they can refer to the article by Rossouw et al in Clinical Chemistry and Laboratory Medicine with regards to Martin-Hopkins performance being more consistent across different analytical platforms.

We mentioned on this article findings in the discussion section. Related paragraph wa given below.

In a previous study made by Rossouw et al., the comparability of direct LDL-C levels obtained Abbott Architect and the Roche Cobas analyzers with the Martin Hopkins, Sampson and Friedewald equations were evaluated, and they found that the performance of predictive equations could be influenced according to the platform and LDL-C levels. Besides, they concluded that Martin/Hopkins formula can be safely implemented for both Abbott and Roche platforms. These findings are in accordance with our findings. 

Please refer to the additional comments on the manuscript PDF.

We thank to the reviewer for his/her comments and suggestions. Relevant corrections were made, taking into account the referee's comments in the relevant document.

---

## [Decision Letter · Decision Letter 1]

27 Apr 2022

Validation of Friedewald, Martin-Hopkins and Sampson Low-Density Lipoprotein Cholesterol Equations

PONE-D-22-02690R1

Dear Dr. Zararsiz,

We’re pleased to inform you that your manuscript has been judged scientifically suitable for publication and will be formally accepted for publication once it meets all outstanding technical requirements.

Kind regards,

Ying-Mei Feng

Academic Editor

PLOS ONE

Additional Editor Comments (optional):

Reviewers' comments:

Reviewer's Responses to Questions

**Comments to the Author**

1. If the authors have adequately addressed your comments raised in a previous round of review and you feel that this manuscript is now acceptable for publication, you may indicate that here to bypass the “Comments to the Author” section, enter your conflict of interest statement in the “Confidential to Editor” section, and submit your "Accept" recommendation.

Reviewer #1: All comments have been addressed

Reviewer #2: All comments have been addressed

Reviewer #3: All comments have been addressed

Reviewer #4: All comments have been addressed

2. Is the manuscript technically sound, and do the data support the conclusions?

Reviewer #1: Yes

Reviewer #2: Yes

Reviewer #3: Yes

Reviewer #4: Yes

3. Has the statistical analysis been performed appropriately and rigorously? 

Reviewer #1: Yes

Reviewer #2: Yes

Reviewer #3: Yes

Reviewer #4: Yes

4. Have the authors made all data underlying the findings in their manuscript fully available?

Reviewer #1: Yes

Reviewer #2: Yes

Reviewer #3: Yes

Reviewer #4: Yes

5. Is the manuscript presented in an intelligible fashion and written in standard English?

Reviewer #1: Yes

Reviewer #2: Yes

Reviewer #3: Yes

Reviewer #4: Yes

6. Review Comments to the Author

Reviewer #1: Zararsız et al analyzed the concordance of various formulas with direct measurement of LDL cholesterol.The better performance of the extended Martin-Hopkins formula was highlighted in this article. If validated for other populations, the application of this new formula will allow better care and treatment in patients at very high cardiovascular risk compared to the most used formula, the Friedewald formula.

Reviewer #2: (No Response)

Reviewer #3: Thank you for the revisions.

My prior comments have been fully addressed.

I do not have any further comments.

Reviewer #4: The author's appear to have addressed all of the reviewer comments and have provided a comprehensive response.

7. PLOS authors have the option to publish the peer review history of their article (what does this mean?). If published, this will include your full peer review and any attached files.

Reviewer #1: No

Reviewer #2: No

Reviewer #3: No

Reviewer #4: No

---

## [Editor Report · Acceptance letter]

6 May 2022

PONE-D-22-02690R1 

Validation of Friedewald, Martin-Hopkins and Sampson Low-Density Lipoprotein Cholesterol Equations 

Dear Dr. Zararsiz:

I'm pleased to inform you that your manuscript has been deemed suitable for publication in PLOS ONE. Congratulations! Your manuscript is now with our production department. 

Kind regards, 

on behalf of

Dr Ying-Mei Feng 

Academic Editor

PLOS ONE